# A Longitudinal Model for a Dynamic Risk Score to Predict Delayed Cerebral Ischemia after Subarachnoid Hemorrhage

**DOI:** 10.3390/bioengineering11100988

**Published:** 2024-09-30

**Authors:** Jan F. Willms, Corinne Inauen, Stefan Yu Bögli, Carl Muroi, Jens M. Boss, Emanuela Keller

**Affiliations:** 1Neurocritical Care Unit, Department of Neurosurgery and Institute of Intensive Care Medicine, Clinical Neuroscience Center, University Hospital Zurich and University of Zurich, 8091 Zurich, Switzerland; neurochirurgie.muroi@hin.ch (C.M.); jensmichael.boss@usz.ch (J.M.B.); emanuela.keller@icucockpit.ch (E.K.); 2Department of Neurology, Clinical Neuroscience Center, University Hospital Zurich and University of Zurich, 8091 Zurich, Switzerland; corinne.inauen@usz.ch (C.I.); stefanyu.boegli@usz.ch (S.Y.B.)

**Keywords:** subarachnoid hemorrhage, brain ischemia, machine learning

## Abstract

Background: Accurate longitudinal risk prediction for DCI (delayed cerebral ischemia) occurrence after subarachnoid hemorrhage (SAH) is essential for clinicians to administer appropriate and timely diagnostics, thereby improving treatment planning and outcome. This study aimed to develop an improved longitudinal DCI prediction model and evaluate its performance in predicting DCI between day 4 and 14 after aneurysm rupture. Methods: Two DCI classification models were trained: (1) a static model based on routinely collected demographics and SAH grading scores and (2) a dynamic model based on results from laboratory and blood gas analysis anchored at the time of DCI. A combined model was derived from these two using a voting approach. Multiple classifiers, including Logistic Regression, Support Vector Machines, Random Forests, Histogram-based Gradient Boosting, and Extremely Randomized Trees, were evaluated through cross-validation using anchored data. A leave-one-out simulation was then performed on the best-performing models to evaluate their longitudinal performance using time-dependent Receiver Operating Characteristic (ROC) analysis. Results: The training dataset included 218 patients, with 89 of them developing DCI (41%). In the anchored ROC analysis, the combined model achieved a ROC AUC of 0.73 ± 0.05 in predicting DCI onset, the static and the dynamic model achieved a ROC AUC of 0.69 ± 0.08 and 0.66 ± 0.08, respectively. In the leave-one-out simulation experiments, the dynamic and voting model showed a highly dynamic risk score (intra-patient score range was 0.25 [0.24, 0.49] and 0.17 [0.12, 0.25] for the dynamic and the voting model, respectively, for DCI occurrence over the course of disease. In the time-dependent ROC analysis, the dynamic model performed best until day 5.4, and afterwards the voting model showed the best performance. Conclusions: A machine learning model for longitudinal DCI risk assessment was developed comprising a static and a dynamic sub-model. The longitudinal performance evaluation highlighted substantial time dependence in model performance, underscoring the need for a longitudinal assessment of prediction models in intensive care settings. Moreover, clinicians need to be aware of these performance variations when performing a risk assessment and weight the different model outputs correspondingly.

## 1. Introduction

### 1.1. Delayed Cerebral Ischemia (DCI) 

DCI occurs in up to one third of patients after aneurysmal subarachnoid hemorrhage [1] and doubles the risk of poor outcome [2]. The pathophysiology of DCI is multifactorial [3] and the prediction and timely detection especially in unconscious patients is highly demanding. 

### 1.2. Tools for the Prediction of DCI

Prediction tools or scores, such as the VASOGRADE [4], currently used in clinical practice are determined on admission and are thus static and do not account for the dynamically changing factors that contribute to the development of DCI. Even in the awake and thus clinically assessable patient, the onset of DCI is missed frequently [5]. In comatose patients, however, clinicians must rely entirely on technical diagnostics such as Transcranial Doppler (TCD), electroencephalography, invasive brain monitoring, CT angiography, CT perfusion, and/or digital subtraction angiography [6]. All of these techniques have their limitations. Some are invasive, examiner dependent, or require patient transport to imaging units. In summary, there are major diagnostic gaps in the DCI time window ranging from day 4 to 14 after initial hemorrhage. 

### 1.3. Optimized Prediction through Machine Learning

Optimized, dynamic, and automated monitoring tools for prediction and detection of DCI are therefore needed. Classical prediction scores for outcome and DCI have already been improved in their performance by linking or adding further parameters and using artificial intelligence (VASOGRADE [4]; SAHIT [7]; Nutshell-tool [8]). Still, most of the available tools fall short in capturing the temporal dynamics of disease progression leading to the occurrence of DCI. Several machine learning models for predicting DCI have already been developed [9]. The majority of these models rely on features collected either immediately after or within the first few days following the cerebral event. These input features are primarily derived from the abovementioned scores, as well as laboratory and physiological parameters. As such, these models are based on single time-point assessments, without accounting for the temporal progression of these features. Consequently, they also fail to capture the individualized and dynamic nature of DCI development. As an exception to be highlighted, Megjhani et al. generated an hourly risk score for DCI development from routine vital signs and analyzed the anchored performance of the models [10]. 

### 1.4. Dynamic Prediction of DCI

Yet, the ability to accurately predict DCI occurrence on a longitudinal scale starting at admission is pivotal for informed clinical decision-making. Thus, we aim to address this gap by developing a longitudinal model taking advantage of static and dynamic clinical parameters and to apply a time-dependent ROC analysis to investigate the varying performance of the algorithm predicting DCI between day 4 and 14. Moreover, we aimed to select machine learning models that provide explainability for medical professionals through the use of visualization techniques.

## 2. Methods

The authors strictly adhered to the TRIPOD (transparent reporting of a multivariable prediction model for individual prognosis or diagnosis) and STROBE (strengthening the reporting of observational studies in epidemiology) reporting guidelines.

### 2.1. Participants and Data Source

For this cohort study, we analyzed data prospectively collected from patients with aneurysmal subarachnoid hemorrhage (aSAH) who were admitted to the neurosurgical intensive care unit at the University Hospital of Zurich between October 2016 and November 2022. The data collection was carried out via a CNS data collector (Moberg ICU Solutions, Ambler, PA, USA) and the high-resolution data were processed and stored by “ICU Cockpit”, our dedicated information technology infrastructure [11]. The study received approval from the ethics committee of Kanton Zurich (Basec Nr. 2016-01101), Switzerland. Patients were included in the study after written consent had been obtained from patients or their legal representatives.

### 2.2. Diagnostics and Treatment

Clinical management was based on the guidelines of the American Heart Association [12,13]. Magnetic resonance imaging (MRI) or computed tomography (CT) scans including CT angiography/perfusion were performed if patients met the following criteria: delayed neurological deterioration (DND) defined as a >2-point change in Glasgow Coma Scale (GCS) or a new focal neurological deficit lasting >1 h and not associated with aneurysmal coiling or clipping [14]. Transcranial Doppler sonography with measurements of mean blood flow velocities were performed daily. In unconscious or sedated patients, multimodal neuromonitoring brain tissue oxygen measurements and cerebral microdialysis were performed. In these patients, a decrease in ptiO2 <20 mmHg and/or an increase in the lactate/pyruvate coefficient above 40 led to one of the abovementioned imaging techniques. Hypertensive therapy was induced with DND and vasospasm in CT-angiography or CT perfusion deficit, respectively, as first line therapy. If patients did not improve or worsened neurologically, or if values from multimodal monitoring did not improve, neuroradiological interventional therapy with intraarterial nimodipine instillation and/or percutaneous balloon angioplasty was performed as second line therapy.

### 2.3. Prediction Target

The outcome, i.e., prediction target, was the first occurrence of delayed cerebral ischemia (DCI). DCI was defined as new infarctions in MRI or CT scans, and/or a confirmed perfusion deficit detected in perfusion CT or MRI between day 4 and day 14 after the onset of symptoms (not present on imaging performed within 24 to 48 h after aneurysm occlusion and not attributable to other causes). We focused only on the first occurrence of DCI excluding any successive instances to avoid the consequent lack of explainability arising from interfering, overlapping DCI symptoms, and predictors.

### 2.4. Predictors

The dataset comprised 30 laboratory findings and 15 values from blood gas analysis, routinely measured in aSAH patients, as well as routinely collected demographics consisting of age, gender, Glasgow Coma Scale, and comorbidities, such as diabetes, cardio-vascular disease, and hypertension. Additionally, standard SAH gradings were considered, including the Hunt and Hess scale, modified Fisher scale, World Federation of Neurosurgical Societies grading system, and the Barrow Neurological Institute grading scale [15,16,17,18]. Data used in this study were collected based on the ICU Cockpit IT infrastructure [11].

### 2.5. Pre-Processing and Missing Data 

In a first step, the longitudinal data of each patient were individually aligned with respect to the time of symptom onset and resampled to a sampling time of one per hour. Forward filling was employed to address gaps in the resampled time series caused by the intermittent data collection of laboratory and blood gas analysis results, thereby always considering the latest value as the current value for each predictor.

Clinical data were acquired prospectively. In case of missing data, during retrospective supplementation of the clinical data, care was taken that patients had a complete set of patient characteristics and SAH gradings where possible. Moreover, by only including commonly measured laboratory findings and blood gas analysis results as longitudinal predictors, missing data could be kept to a minimum. Nonetheless, to allow for incomplete predictor vectors also at time of prediction, all model pipelines evaluated in this study included a simple median imputer as a pre-processing step. 

### 2.6. Data Modeling

The modeling task was divided into two sub-tasks: modeling predictors that remained constant after recording at admission and time-varying predictors that changed during the ICU stay (i.e., laboratory findings, blood gas analysis results, with age as the only exception). Separate groups of models were developed for each task and subsequently evaluated using 10-fold nested cross-validation. These two groups of models are referred to as static and dynamic models, respectively. The cross-validation folds were stratified to ensure equal DCI prevalence, and the stratification grouped the data in such a way that information from a specific patient appeared only in a single fold. The primary metric used for model evaluation was the area under the receiver operating characteristics curve (ROC AUC), which was also utilized in the optimization procedures during model training. 

For both groups of models, the modeling task was framed as a classification problem. For static models, the task was to distinguish between patients that eventually experienced a DCI from those that did not. For the dynamic models, the goal was to differentiate between patients that would or would not experience a DCI during the subsequent 48 h. For the latter, we used the first DCI event as an anchor following an approach presented by Megjhani et al. [10]. For patients with no DCI event, the anchor was set to the median time between symptom onset and first DCI occurrence, the latter calculated from patients with DCI. Importantly, for each patient, we included all 48 data points (hourly predictor vectors) leading up to the anchor in the development dataset. This was done to account for the varying degree of variation in the dynamic predictor values over the considered 48 h time window before the anchors. 

The evaluated static models, i.e., models trained only on static predictors, comprised Logistic Regression models with L1 and L2 penalty, Support Vector Classifier, Random Forests, Extremely Randomized Trees, and Histogram-based Gradient Boosting Classification Tree. For the Logistic Regression and Support Vector Classifier models, the predictors were centered around zero and scaled to have unit variance. Class weights were adjusted in ensemble models to balance the dataset. Furthermore, hyperparameter tuning was performed and sequential feature selection was evaluated in comparison with a set of baseline models based on default parameters and all predictors.

The dynamic models included Logistic Regression models, Extremely Randomized Trees, and Histogram-based Gradient Boosting Classification Tree. As optimization routes, we explored Principal Component Analysis (PCA), Feature Selection based on Gini-Importance (FS from Model), and Recurrent Feature Elimination (RFE) in combination with hyperparameter tuning. Again, class weights were adjusted in training ensemble models to balance the dataset.

Finally, we also evaluated a model combining the static and the dynamic models in a voting model. The voting model computed its output score as the arithmetic average of the static and dynamic scores and was not separately trained. 

The data modelling was performed using Python 3.9.12 and the machine learning module scikit-learn version 1.1.1.

### 2.7. Leave-One-Out Simulation

In addition to the anchored ROC analysis, leave-one-out (LOO) simulation was conducted to evaluate the model’s ability to predict DCI in a scenario more closely resembling the clinical setting in which the models would regularly produce new risk assessments. Due to the considerable amount of time required to train all the different evaluated classifiers, we restricted the LOO simulation analysis to the best performing models in the anchored analysis. For each run of the LOO simulation, a single patient was excluded from the development set and the models were re-trained on the data of the remaining patients. We then used these newly trained models to predict hourly risk scores for the left-out patient for their entire ICU stay. The procedure was repeated until we had computed hourly risk scores for all patients. Finally, time-dependent ROC analyses were performed to evaluate and summarize the performance of the models trained on the leave-one-out datasets. 

In a first approach, the overall ability of the models to differentiate between the two groups of patients with and without eventual DCI occurrence was studied. The output scores of the models were plotted for the two groups together with the time-dependent ROC AUC values. 

Next, the focus was shifted from classification regarding eventual DCI occurrence towards DCI occurrence in the following 48 h. The latter being arguably the clinically more relevant question to be answered by a prediction tool. To this end, we evaluated whether the scores during the 48 h period before a DCI occurrence were significantly higher compared to the periods before or in patients with no DCI at all. This comparison was first carried out by pooling all the hourly risk scores from the entire ICU stays and a second time in a time-resolved manner in order to study the performance of the prediction model as function of time starting with the time of symptom onset. For the latter, the considered time frame from day 2 to day 14 was divided into 5 intervals of equal proportions of time-points that were followed by a DCI. 

Importantly, when analyzing the predictions for the next 48 h, the risk set, i.e., the patients being assessed, was continuously adjusted to contain solely patients who had not experienced a DCI event. In other words, once patients suffered from a DCI, they were excluded from the set of analyzed patients. 

### 2.8. Statistical Analysis

The Mann–Whitney U-test was used for comparing continuous and ordinal variables and the Fisher exact test for binary variables. Statistical significance was assumed at *p* < 0.05. Effect size was quantified using the ROC AUC value, where a value above 0.5 indicates a positive association and below 0.5 a negative association with the outcome. Confidence intervals for the ROC AUC values in the time-dependent ROC analysis were computed via bootstrapping.

## 3. Results

### 3.1. Participants

In total, the data of 222 patients were analyzed. Of these patients, 218 patients had data in the relevant time frame for DCI occurrence between day 4 and 14 and were included in the development set. DCI occurred in 89 (41%) of patients. An overview of the patient characteristics is shown in Table 1. With a median age of 57 years and a higher proportion of females (63.8%), this cohort is representative of typical subarachnoid hemorrhage patients. The severity of subarachnoid hemorrhage, as assessed by clinical scales (WFNS, Hunt and Hess), showed no significant association with the development of DCI in our cohort. In contrast, the severity classification based on imaging findings (mFS, BNI) yielded different results. Statistical analysis revealed that only the Barrow Neurological Institute (BNI) grading scale was significantly associated with DCI occurrence. 

### 3.2. Model Development and Model Specification

From the 218 patients in the development set, 10,470 predictor vectors were sampled for the 48 h before the anchor, each containing a total of 60 static and dynamic parameters. On average, 3.6% and 7.2% of the static and dynamic parameters were missing, respectively. The dataset was sightly unbalanced with 41% of the patients experiencing at least one DCI event. 14 static and 19 dynamic model pipelines of different complexity were defined and evaluated. 

### 3.3. Performance of Model Building Pipelines (Cross-Validation)

The results of the cross-validation results are presented and summarized in Figure 1. 

Among the static models, hyper-parameter tuning improved model performance across all models. The simple Logistic Regression models outperformed the more complex machine learning models. Specifically, the Logistic Regression model with Sequential Feature Selection and the L2 loss-function showed the best performance (ROC AUC: 0.66 ± 0.08). When trained on the entire development set, this pipeline selected the BNI grading scale, the WFNS, diabetes, hypertension, and cardiovascular disease as predictors, along with a high regularization parameter (C = 10) during hyper-parameter tuning. 

Regarding the dynamic models, the baseline Extremely Randomized Trees (ET) model achieved the best performance (ROC AUC: 0.70 ± 0.09), while the ET GS model with hyper-parameter tuning ranked second (ROC AUC: 0.69 ± 0.08). Consequently, the ET GS model was used in the combined model as well as in subsequent leave-one-out analysis, as it shows a similar performance as the ET baseline model, while having its model parameters learnt from the data. When trained on the entire development set, the optimized parameters of the ET GS model for the maximal tree depth and the number of selectable features at the decision nodes were determined as 8 and 10, respectively. 

Using the best static and the selected optimized dynamic model in a combined model resulted in a superior model achieving a ROC AUC of 0.73 ± 0.05. 

Figure 2 depicts the corresponding ROC curves of the selected static and dynamic models, as well as the resulting combined model. The ROC curve computations were based on the cross-validation scores, thus, represent an average over the individual models fitted during cross-validation.

### 3.4. Model Performance (Leave-One-Out)

Figure 3 shows the scores of the selected static, dynamic and the resulting combined models as computed in the LOO simulation. While the output of the static model naturally stayed constant over the entire stay, with scores higher for the patients in the DCI group, the outputs of the dynamic and combined model changed continuously. After day 2, the scores of the dynamic DCI group visibly surpass the ones of the static DCI group and show a ROC AUC value above 0.5, further increasing until day 6 when it starts saturating at around 0.67, reaching a maximum of 0.715 on day 10. 

When analyzing the performance of DCI prediction for the following 48 h, we found that the risk scores were significantly higher for all three models during the 48 h leading up to a DCI occurrence compared to when no DCI occurred. Corresponding boxplots are shown in Figure 4. The effect sizes measured by the ROC AUC were 0.60, 0.61, and 0.63 for the static, dynamic, and combined model, respectively.

To study the time-dependence of the model performances, five time-intervals were computed with equal numbers of samples (e.g., hours) followed by a DCI during subsequent 48 h. The intervals were (i) day 2 until day 4.2, (ii) day 4.2 until day 5.4, (iii) day 5.4 until day 6.9, (iv) day 6.9 until day 8.8, and (v) day 8.8 until day 14. The different lengths stemmed from the uneven distribution of DCI occurrences after SAH. The ROC AUC values for all three models are shown in Figure 5. The static model showed the strongest time dependence with a ROC AUC that was below 0.5 in the first interval. The dynamic model is more stable, achieving ROC AUC values between 0.62 and 0.67. The dynamic model is the best-performing model for the first two intervals until day 5.4. From day 5.4 onwards, the combined model shows the best performances with ROC AUC values of 0.68, 0.67, and 0.71 for the third, fourth, and fifth intervals, respectively.

To make the scores more interpretable, thresholds were computed that achieve a sensitivity level of 80%. Table 2 presents these thresholds along with the corresponding specificity for each interval. The thresholds depend on the considered time intervals and increase together with the corresponding ROC AUC value. 

## 4. Discussion

Numerous scoring systems exist for risk stratification in aSAH patients. However, a critical need persists for continuously updating clinical support tools that enhance situational awareness. Recently, Megjhani et al. presented a set of machine learning models trained on vital sign parameters and patient demographics using DCI occurrence as an anchor to train binary classifiers [10]. While their models demonstrated robust performance when applied to other centers, their analysis was limited to anchored data. In addition, there is the ongoing problem that models developed are not validated on live data streams. These limitations hinder the assessment of their applicability in real-world clinical practice, where the anchor is unknown, and data are aligned with symptom onset or hospital/ICU admission.

Adopting the anchoring approach in building our development dataset, we trained well-performing static and dynamic models evaluating a range of machine learning model types. Furthermore, disentangling static and dynamic parameters allowed us to come up with a more transparent combined model that permits reasoning about the different model contributions.

To justify the anchoring approach used and to evaluate the resulting models within a scenario closely resembling clinical practice, we conducted time-dependent ROC analysis using scores generated through a leave-one-out simulation. This allowed us to investigate the performance not anchored to the DCI events anymore but to symptom onset, which is what a clinician would experience in clinical practice. Our time-dependent analysis of the scores showed that the dynamic model does indeed produce highly time-dependent results. Interestingly, when grouping patients by eventual DCI occurrence, the median dynamic score initially trended even higher in the patients without eventual DCI before declining below the DCI patients around day 4, indicating that the correlations learned by the dynamic model from the anchored dataset are not indicative for DCI in the early phase of the patient’s ICU stay. 

Moreover, focusing on the 48 h before DCI onset, we confirmed significantly higher scores for all models compared to other periods. Additionally, partitioning the time interval up to day 14 into five intervals revealed that model performance strongly depends on the duration from symptom onset. While temporal variation in the static model’s performance is influenced by changes in the risk set, the dynamic model’s performance is governed by disease progression and corresponding fluctuations in laboratory and blood gas analysis results.

These temporal variations in model performance underscore the challenges associated with longitudinal models, including the need to adapt threshold interpretation over time to maintain consistent sensitivity. Furthermore, selecting the appropriate model or sub-model is crucial for achieving optimal risk assessment. The recommended strategy is to utilize the dynamic model until day 5.4 and then switch to the combined model.

A key objective for our team is to obtain direct clinical experience with our support system. To facilitate this, we deliberately selected models that allow for visualization, thereby offering a degree of interpretability for medical personnel. Figure 6 demonstrates the implementation of the visualization tool. A waterfall plot illustrates the influence of individual parameters on the static score, while a heat map depicts the impact of laboratory chemical values on the dynamic score. Additionally, the combined score can be visualized over time using a separate graph. In a future clinical study, the goal will be to to equip medical staff with interpretative tools that may foster trust in the algorithms.

However, this study has several limitations, including the small sample size and data collected at a single center. Given the limited sample size, we opted for a simple voting approach instead of learning model combination weights from the data. Additionally, the lack of data from multiple centers prevents us from investigating the transferability of our models to different healthcare settings.

We only implemented laboratory parameters and values from the blood gas analyses in our dynamic model. Several considerations were made prior to this decision. As early as 2010, Kasius et al. were able to show a correlation between an increase in platelet and leucocyte counts and the occurrence of DCI [19]. Other correlations, for example via serum D-dimers or C-reactive protein, have also followed in more recent publications [20]. A further reason for excluding vital parameters or measured values of extended diagnostics is based on the consideration that these imply a high degree of false measurements. This may be due to artefacts or interventionally altered values as a result of specific treatment. In addition, laboratory and blood gas analyses are ubiquitously available, even in smaller hospitals or in countries with less data processing power. 

The definition of delayed cerebral ischemia as an endpoint remains difficult, especially in unconscious patients, where clinical examination is limited. We therefore confirmed the endpoint in all patients with imaging evidence of new infarction and/or evidence of perfusion deficit. From a clinical point of view, it is therefore not possible to correctly determine the time of DCI onset, especially in unconscious patients, which affects the time frame of the dynamic model in both training and validation.

## 5. Conclusions

A longitudinal DCI prediction model with static and dynamic sub-models could be trained successfully and validated in a real-world scenario. The time-dependent ROC analysis revealed that the performance of the sub-models as well as the combined model critically depended on the time of assessment. This highlighted the importance to evaluate longitudinal prediction models using time-dependent analysis methods such as time-dependent ROC analysis. This holds especially true for prediction targets that evolve over time. The next important steps are to validate our model in external datasets and in everyday clinical practice using live data streams.

## Figures and Tables

**Figure 1 bioengineering-11-00988-f001:**
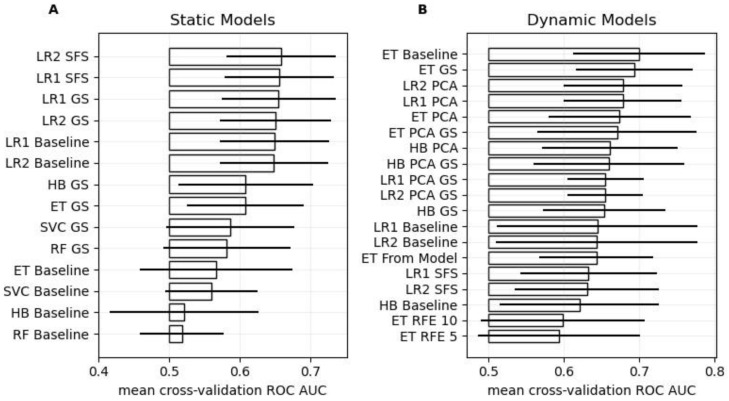
Cross-validation performance. Panels (**A**,**B**) show the ROC AUC scores for the different static and dynamic models, respectively. The bars and the horizontal lines indicate the mean and standard deviation of ROC AUC score over the 10 folds, respectively. The baseline models were parameterized with reasonable default values, but no hyper-parameter tuning, feature selection, or dimensionality reduction was performed. GS stands for Grid Search (i.e., hyper-parameter tuning), PCA for Principal Component Analysis (i.e., dimensionality reduction), SFS for Sequential Feature Selection, RFE for Recursive Feature Elimination, and “From Model” implies that 10 features were selected using feature importance derived from said model. Models are abbreviated as follows: Extremely Randomized Trees (ET), Histogram-based Gradient Boosting Classification Tree (HB), Random Forest (RF), Logistic Regression with L1 or L2 loss (LR1 or LR2, respectively), and Support Vector Machine (SVC).

**Figure 2 bioengineering-11-00988-f002:**
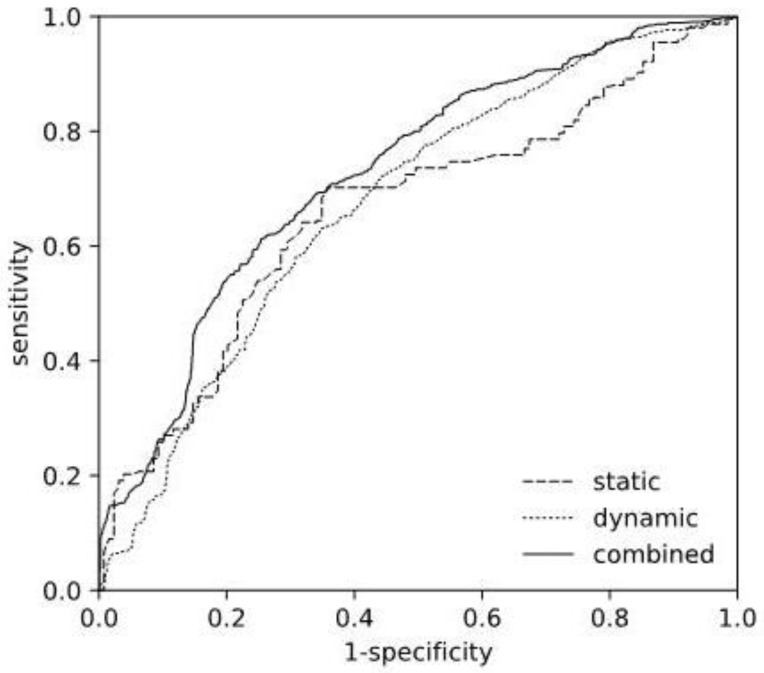
Receiver Operating Characteristics Curve. Comparison of ROC curves of the selected static (L2 SFS), the dynamic (ET GS) models, and the resulting combined model.

**Figure 3 bioengineering-11-00988-f003:**
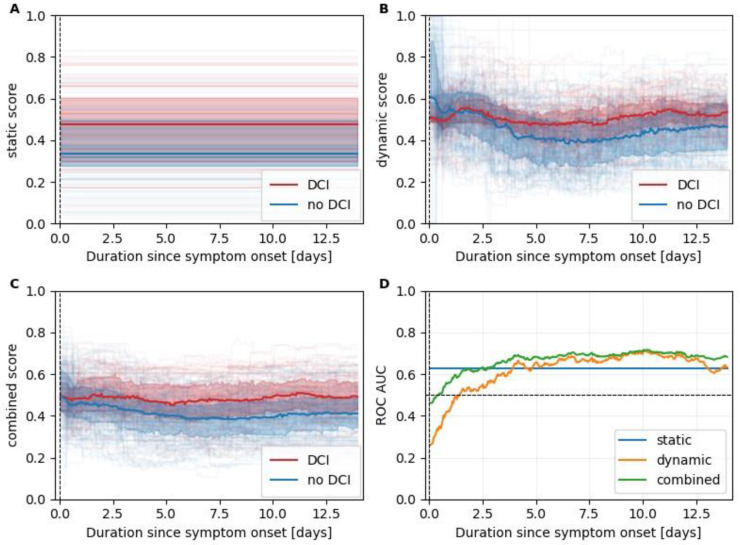
Leave-one-out scores. Panels (**A**–**C**) summarize leave-one-out scores for the static, dynamic, and combined models, respectively. Scores of patients with and without an eventual DCI are plotted in red and blue, respectively. The median scores are plotted as a line together with the interquartile range (colored area). In the background thin, partially transparent lines indicate the scores of individual patients. In panel (**D**), we plotted the time-dependent ROC AUC values to identify patients exhibiting at least one DCI occurrence.

**Figure 4 bioengineering-11-00988-f004:**
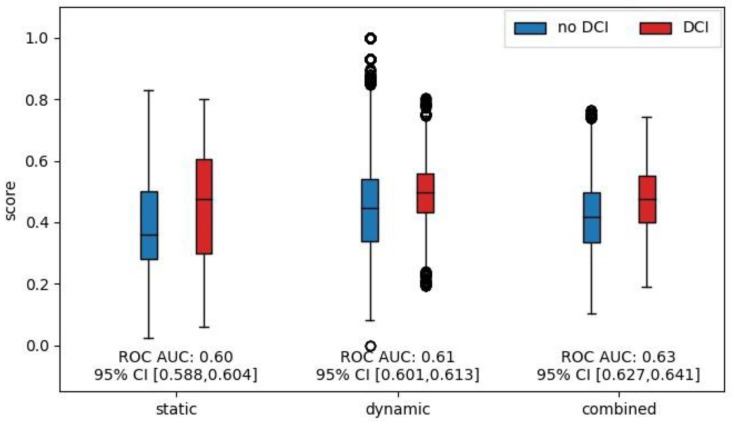
Comparison of model output scores for the 48 h leading up to DCI. Boxplots of model output scores for the different models. Scores were grouped according to whether a DCI occurred in the subsequent 48 h. Only scores up to the first DCI were considered and patients were removed from the risk set after DCI occurrence. The effect size was measured and is shown as the ROC AUC.

**Figure 5 bioengineering-11-00988-f005:**
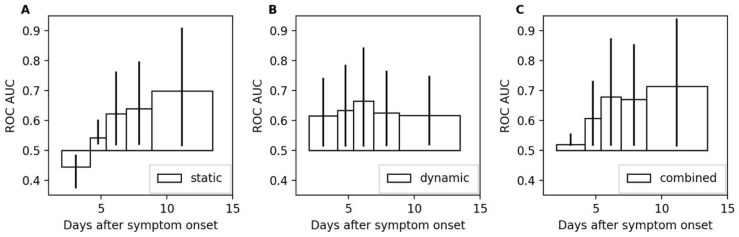
Area under the receiver operating characteristic for different intervals. Panels (**A**–**C**) show the ROC AUC values computed for the intervals (i) day 2 until day 4.2, (ii) day 4.2 until day 5.4, (iii) day 5.4 until day 6.9, (iv) day 6.9 until day 8.8, and (v) day 8.8 until day 14 after symptom onset. The 95% confidence intervals are depicted by vertical lines at the tips of the bars.

**Figure 6 bioengineering-11-00988-f006:**
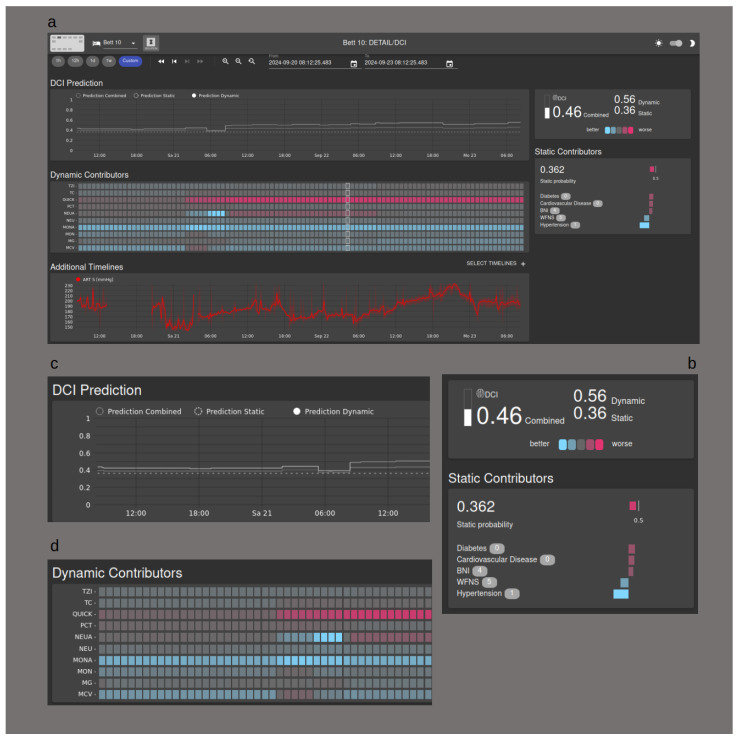
Visualization Tool. (**a**) Overview of the user interface; (**b**) the influence of the static contributors is shown as a waterfall chart; (**c**) the DCI probability is shown as a graph over time; (**d**) the dynamic contributors are shown in a heat map.

**Table 1 bioengineering-11-00988-t001:** Patient characteristics for patients with and without DCI. GCS (Glasgow Coma Scale), WFNS (World Federation of Neurosurgical Societies), mFS (modified Fisher Scale), and BNI (Barrow Neurological Institute grading scale). * Significant.

Total Patients = 218		DCI+	DCI−	*p* Value
Age, year; median (IQR)	57.0 (50.0–67.0)	56.0 (49.0–64.0)	57.0 (50.0–69.0)	0.289
Female sex, *n* (%)	139 (63.8)	56 (62.9)	83 (64.3)	0.886
Hypertension, *n* (%)	83 (38.1)	28 (31.5)	55 (42.6)	0.118
Cardiovascular disease, *n* (%)	43 (19.7)	13 (14.6)	30 (23.3)	0.123
Diabetes, *n* (%)	17 (7.8)	3 (3.4)	14 (10.9)	0.069
GCS (extra.), median (IQR)	14.0 (8.0–15.0)	14.0 (9.0–15.0)	14.0 (8.0–15.0)	0.959
Hunt and Hess, median (IQR)	3.0 (2.0–4.0)	3.0 (2.0–4.0)	2.0 (2.0–4.0)	0.103
Hunt and Hess, 4–5; *n* (%)	58 (26.6)	25 (28.1)	33 (25.6)	0.756
WFNS, median (IQR)	2.0 (1.0–4.0)	2.0 (1.0–4.0)	2.0 (1.0–4.0)	0.972
WFNS, 4–5; *n* (%)	91 (41.7)	35 (39.3)	56 (43.4)	0.578
mFS, median (IQR)	4.0 (3.0–4.0)	4.0 (3.0–4.0)	4.0 (3.0–4.0)	0.157
mFS, 3–4; *n* (%)	199 (91.3)	84 (94.4)	115 (89.1)	0.225
BNI, median (IQR)	3.0 (3.0–4.0)	4.0 (3.0–5.0)	3.0 (2.0–4.0)	<0.001 *
BNI, 4–5; *n* (%)	98 (45.0)	52 (58.4)	46 (35.7)	0.001 *

**Table 2 bioengineering-11-00988-t002:** Thresholds and specificity for a sensitivity level of 80%.

Model	Interval	(i)	(ii)	(iii)	(iv)	(v)
static	ROC AUC	0.44	0.54	0.62	0.64	0.70
	Threshold	0.21	0.26	0.30	0.30	0.35
	Specificity	0.13	0.19	0.30	0.32	0.49
	Precision	0.07	0.15	0.15	0.14	0.09
dynamic	ROC AUC	0.62	0.63	0.67	0.63	0.62
	Threshold	0.46	0.41	0.39	0.38	0.41
	Specificity	0.37	0.45	0.45	0.44	0.40
	Precision	0.09	0.20	0.18	0.16	0.08
combined	ROC AUC	0.52	0.61	0.68	0.67	0.71
	Threshold	0.38	0.36	0.37	0.39	0.42
	Specificity	0.26	0.29	0.37	0.47	0.55
	Precision	0.08	0.16	0.16	0.17	0.10

## Data Availability

All data generated or analyzed during the current study are available from the corresponding author on reasonable request.

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
