# Peer review of "A Longitudinal Model for a Dynamic Risk Score to Predict Delayed Cerebral Ischemia after Subarachnoid Hemorrhage"

_bioengineering, 2024, doi:10.3390/bioengineering11100988_

Round 1

Reviewer 1 Report

Comments and Suggestions for Authors

span lang="EN-US" Jan F. Willms em style="mso-bidi-font-style: normal"et al/em. reported an interesting work about a prediction model for DCI after SAH. The topic was less frequently discussed yet important. It was of a certain significance nowadays. The reviewer suggested a Minor Revision for this submission. Please refer to the following comments./span/p p class="MsoListParagraph" style="margin-left: 18.0pt; text-indent: -18.0pt; mso-char-indent-count: 0; mso-list: l0 level1 lfo1"1-      Please divide the Introduction Section into several paragraphs, and state the significance of this work in the last paragraph.

2-      There were a lots of parameters without significant difference in Table 1, but they should also be briefly discussed in Section 3.1.

3-      How to obtain the threshold values in Table 2? Were they trustful? Please make proper comments.

4-      More screencaptures of the UI for visualization tools could be shown in Figure 6.

5-      Please delete the “Instruction for authors” at the end of Conclusion Section.

Author Response

Thank you for the opportunity to resubmit our revised manuscript to Bioengineering. Based on the Reviewers comments, the quality of the publication could be improved significantly. Please find our point-by-point response to the Reviewers below:

Reviewer #1

The study group reported an interesting work about a prediction model for DCI after SAH. The topic was less frequently discussed yet important. It was of a certain significance nowadays. The reviewer suggested a Minor Revision for this submission. Please refer to the following comments:

1-      Please divide the Introduction Section into several paragraphs, and state the significance of this work in the last paragraph.

Thank you for improving the structure and clarifying our purpose. We have formed paragraphs in the introduction with headings and focused the conclusions on major findings (lines 39-79). In the same step, we also focused on the conclusions at the end of the manuscript (lines 402-403).

2-      There were a lots of parameters without significant difference in Table 1, but they should also be briefly discussed in Section 3.1.

That is a useful suggestion, thank you. We have revised section 3.1 and briefly interpreted the table (lines 224-230)

3-      How to obtain the threshold values in Table 2? Were they trustful? Please make proper comments.

Thank you very much for this valid comment. We agree that the original passage could have been clearer. In response, we have reformulated the passage to emphasize that the thresholds were computed specifically to achieve a sensitivity level of 80% in each time interval. (lines 324-326)

4-      More screen captures of the UI for visualization tools could be shown in Figure 6.

Thank you for this valuable suggestion. We have added additional screenshots. (lines 376-379)

5-      Please delete the “Instruction for authors” at the end of Conclusion Section.

We have deleted this in the generated Word document.

Reviewer 2 Report

Comments and Suggestions for Authors

Overall, it is an interesting study to predict delayed cerebral ischemia. Here are some minor comments, which should be addressed before the publication.

1.        Author introduces the “ROC” (line 23) in the abstract, however, there is no abbreviation mentioned on the first page.

2.        Author did mention in the introduction parts “Several machine learning models for predicting DCI have already been developed”.  How are these methods not good enough for accurate prediction? There is no proper description in the introduction. Author should enlarge the introduction parts.

3.        I don’t think, it is well justified to use the word “novel” in the “novel longitudinal DCI prediction model” mentioned in the abstract section. This is an improved longitudinal DCI prediction model.

4.        I believe it will be interesting to see the validation of the current model in other data sets in future studies.

Author Response

Reviewer #2:

Overall, it is an interesting study to predict delayed cerebral ischemia. Here are some minor comments, which should be addressed before the publication.

  1. Author introduces the “ROC” (line 23) in the abstract, however, there is no abbreviation mentioned on the first page.

Thanks for the advice. We have added an explanation of the abbreviation (lines 22-24).

  1. Author did mention in the introduction parts “Several machine learning models for predicting DCI have already been developed”. How are these methods not good enough for accurate prediction? There is no proper description in the introduction. Author should enlarge the introduction parts.

This is an important comment. We cannot discuss all points of the individual publications, but we have included a brief explanation (lines 63-69).

  1. I don’t think, it is well justified to use the word “novel” in the “novel longitudinal DCI prediction model” mentioned in the abstract section. This is an improved longitudinal DCI prediction model.

Thank you very much for the recommendation, which we are happy to accept as it is better suited to our project. (line 15)

  1. I believe it will be interesting to see the validation of the current model in other data sets in future studies.

We are currently working on improving our model with additional features, also with the aim of validating it on “external” data sets.